# Designable ultra-smooth ultra-thin solid-electrolyte interphases of three alkali metal anodes

Yu Gu[1], Wei-Wei Wang[1], Yi-Juan Li[1], Qi-Hui Wu[2], Shuai Tang[1], Jia-Wei Yan[1], Ming-Sen Zheng [1], De-Yin Wu[1], Chun-Hai Fan [3], Wei-Qiang Hu[1], Zhao-Bin Chen[1], Yuan Fang[1], Qing-Hong Zhang[1], Quan-Feng Dong[1] & Bing-Wei Mao[1]

Dendrite growth of alkali metal anodes limited their lifetime for charge/discharge cycling. Here, we report near-perfect anodes of lithium, sodium, and potassium metals achieved by electrochemical polishing, which removes microscopic defects and creates ultra-smooth ultra-thin solid-electrolyte interphase layers at metal surfaces for providing a homogeneous environment. Precise characterizations by AFM force probing with corroborative in-depth XPS profile analysis reveal that the ultra-smooth ultra-thin solid-electrolyte interphase can be designed to have alternating inorganic-rich and organic-rich/mixed multi-layered structure, which offers mechanical property of coupled rigidity and elasticity. The polished metal anodes exhibit significantly enhanced cycling stability, specifically the lithium anodes can cycle for over 200 times at a real current density of 2 mA cm$^{-2}$ with 100% depth of discharge. Our work illustrates that an ultra-smooth ultra-thin solid-electrolyte interphase may be robust enough to suppress dendrite growth and thus serve as an initial layer for further improved protection of alkali metal anodes.

---

[1] State Key Laboratory of Physical Chemistry of Solid Surfaces and Department of Chemistry, iChEM, College of Chemistry and Chemical Engineering, Xiamen University, Xiamen 361005, China. [2] Department of Materials Chemistry, College of Chemical Engineering and Materials Science, Quanzhou Normal University, Quanzhou 362000, China. [3] Division of Physical Biology & Bioimaging Center, Shanghai Synchrotron Radiation Facility, CAS Key Laboratory of Interfacial Physics and Technology, Shanghai Institute of Applied Physics, Chinese Academy of Sciences, Shanghai 201800, China. Correspondence and requests for materials should be addressed to Q.-F.D. (email: qfdong@xmu.edu.cn) or to B.-W.M. (email: bwmao@xmu.edu.cn)

Lithium anode has received considerable attention, as it is regarded as the most promising anode candidate for the next generation high-energy-density rechargeable batteries such as Li–sulfur and Li–air batteries[1–6]. Recently, sodium and potassium anodes are also gaining popularity because of their abundance and more attractive cathode chemistry in Na–air[7] and K–air[8] batteries. However, anodes made of these alkali metals, also denoted as $M_A$, suffer from intrinsic and induced dendrite growth upon charge–discharge cycling, resulting in low Coulombic efficiency, short-circuiting and thus short lifetime of batteries[9–13]. Continuous efforts have been devoted to improve the long-term stability of Li anodes, ranging from traditional soaking-based method that passively forms a solid-electrolyte interphase (SEI)[14–19], strategies to improve Li deposition/dissolution behaviors including preferential adsorption[20], employment of ultrahigh Li salt concentration[21] and asymmetric cycling protocols[22], to currently overwhelming artificial approaches[23–28] that form micrometer scale artificial SEI layers by heavy reactions with Li surface or application of physical isolating layers. However, up to date, the long-term stability of Li anodes, especially under high-current density and with reasonable depth of discharge (DOD), is still far from satisfactory (see Supplementary Table 1). Most of the works have to employ Li foils as anode material because of low utilization of Li source, or only limited cycle numbers can be maintained in the case of anode-free type cells. The situation is even less optimistic for Na and K anodes[29,30] (see Supplementary Table 2).

Looking into the origins of dendrite growth, the microscopic protrusions, non-uniform electric field distribution, as well as uneven supply of metal ion flux are detriment factors that promote dendrite growth[2,3]. These factors are inherently correlated with the roughness of surface as well as inhomogeneity of SEI that is inevitably formed on the $M_A$ surfaces as a result of chemical and/or electrochemical reductions of electrolyte[14,31–33]. The rough surface morphology boosts the intrinsic growth of long dendrites at Li anode, while inhomogeneous and unstable SEI induces dendrite growth at all of the three metal alkali anodes,

especially Na and K anodes because of their more reactive chemistry than Li (Fig. 1a). However, it is important to emphasize that SEI is a double-edged sword depending on its physical and chemical properties: A coarse and inhomogeneous SEI, such as the disordered mosaic type of SEI prepared by soaking-based method[34], promotes preferential growth through the cracks of mosaic SEI or at the locations where SEI is thinner or broken; while a fine and smooth SEI where the localized defects are largely eliminated would suppress both intrinsic and induced dendrite growth of all the three $M_A$ anodes.

Ideally, SEI layers for alkali metal anodes must be chemically stable and metal ion-conductive for batteries to operate; they should also be compact in general and uniform laterally to avoid localized effects for dendrite growth; last but not least, they should have well-defined structure both in lateral and vertical directions with mechanical property of coupled rigidity and elasticity to accommodate the volume change upon cycling[2,35]. Of particular attention is that the achievement of such ideal SEIs relies strongly on the smoothness of the metal surface[19,36]. Thus, creation of atomically smooth surface, onto which a near-ideal SEI can be formed, is an ultimate goal that would lead to a near-perfect metal alkali anode. This has been pursued for long time, but is so challenging that has not been achieved by currently available approaches. However, it is well-known that electrolytes can be reduced to various oxidation states[37,38] depending on potential and $M_A^+$ concentration. Hence, the fundamental surface electrochemistry could play important and unique roles that other approaches cannot provide in terms of facile control of electrode kinetics.

Here, we report a general non-conventional electrochemical approach to create near-perfect Li, Na, and K metal anodes, based on electrochemical polishing of the alkali metal surfaces down to atomic-flatness as well as manipulation of electrolyte reduction processes to construct ultra-smooth ultra-thin (USUT) SEI with designable structure. By combined extensive characterizations by chemically, structurally as well as mechanically sensitive techniques, we reveal that the SEIs on $M_A$ anodes can be facilely

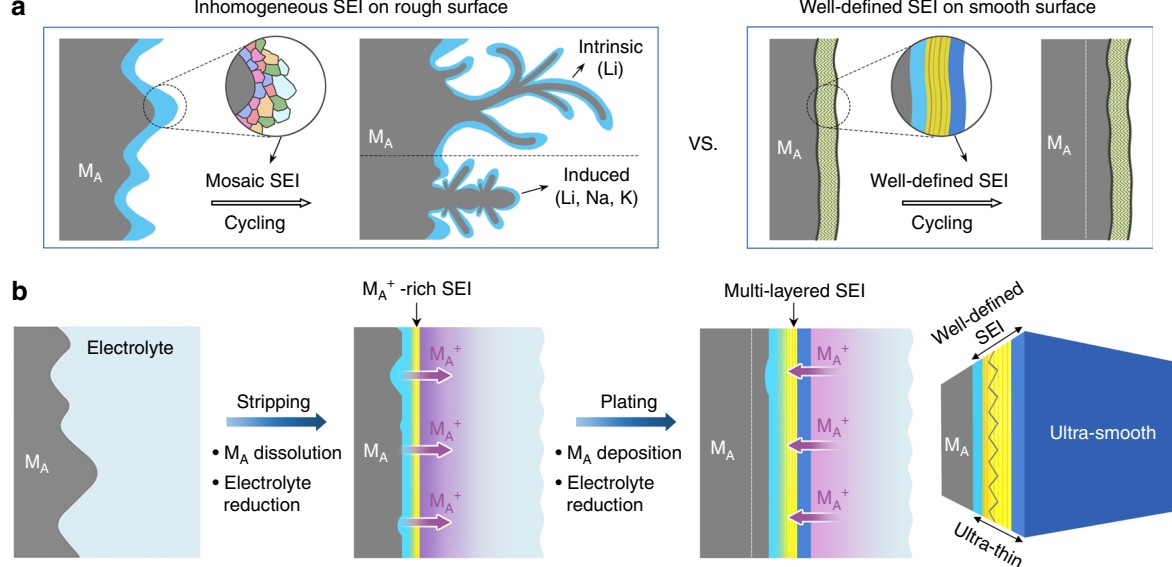

**Fig. 1** Schematic diagrams of dendrites growth on different $M_A$ surfaces and electrochemical stripping-plating strategy for polishing $M_A$ surface. **a** Schematic illustration of cycling process on a rough surface with inhomogeneous SEI and a smooth surface with well-defined SEI. **b** Scheme of the potentiostatic stripping and galvanostatic plating for polishing of and formation of SEI on $M_A$ surface. The stripping step involves concurrent $M_A$ dissolution and electrolyte reduction while the plating step enables $M_A$ back deposition and further electrolyte reduction. The layers in light blue and dark blue represent the inorganic-rich inner layer and inorganic-rich surface layer of SEI, respectively, while the layer in yellow color represents the organic-rich middle layer

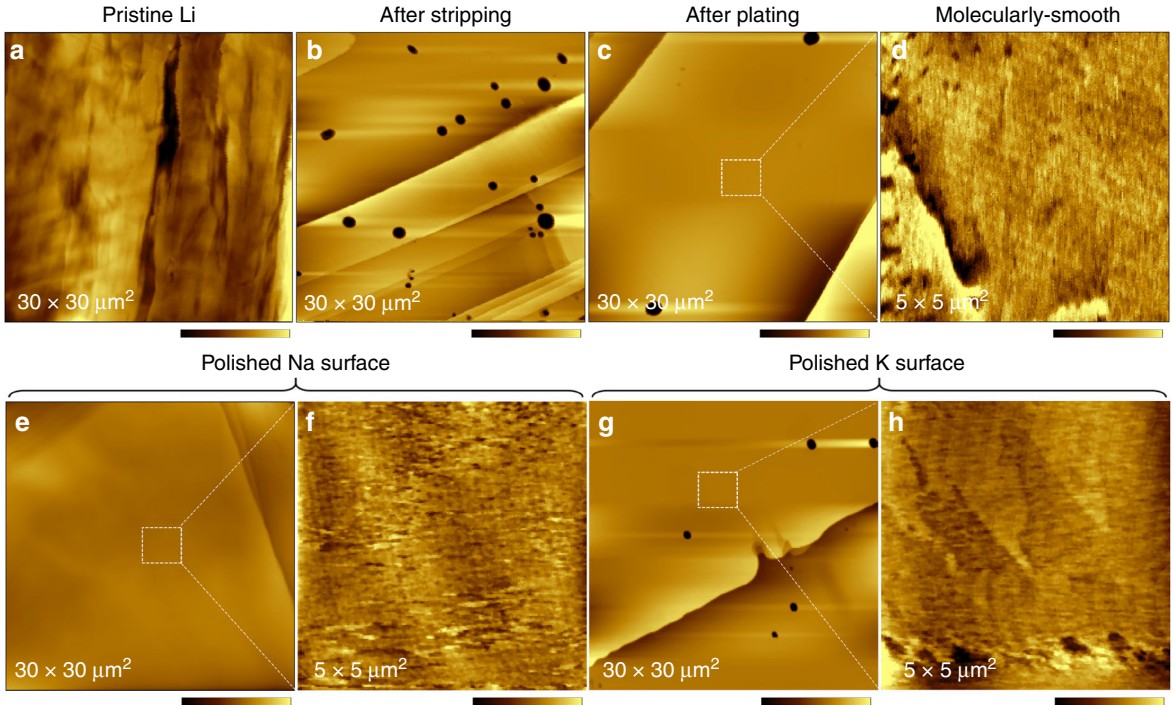

**Fig. 2** AFM characterization of morphology of polished $M_A$ surfaces. **a–d** AFM images of Li surface before (**a**) and after stripping (**b**) and plating (**c, d**) in the electrolyte of 1 M LiTFSI/DME-DOL. **e, f** AFM images of Na surface after polishing in the electrolyte of 1 M NaOTf/diglyme. **g, h** AFM images of K surface after polishing in the electrolyte of 1 M KTFSI/DME. Color bars are 0–1 μm (**a**), 0–300 nm (**b, c**), 0–1.5 nm (**d**), 0–150 nm (**e**), 0–200 nm (**g**), and 0–1.5 nm (**f, h**)

tuned to have alternating inorganic-rich (I) and organic-rich/mixed (O) multi-layer structures in appropriate electrolytes. These types of multi-layer SEI structures are on contrary to the traditional mosaic model of SEI, and bear mechanical property of coupled rigidity and elasticity. With these features and merits, a significant step forward has been achieved for all the three alkali metal anodes in cycling stability under high-current density with high DOD.

## Results

**Electrochemical polishing for alkali metal surfaces**. Basically, our electrochemical approach is based on the electrochemical stripping-plating (ESP) strategy as shown in Fig. 1b, including a potentiostatic stripping, during which concurrent $M_A$ dissolution and electrolyte reduction takes place, and a follow-up galvanostatic plating, during which concurrent $M_A$ deposition and further reduction of the electrolyte occurs. The potential for stripping is sufficiently anodic for high-rate $M_A$ dissolution (up to ~200 mA cm$^{-2}$ in the initial stage), yet within the potential window for electrolyte reduction. An exceedingly high $M_A^+$ surface concentration (~11 mol L$^{-1}$) is created during stripping (see Supplementary Fig. 1, Supplementary Notes 1 and 2 and Supplementary Table 3), which forms a viscous $M_A^+$ liquid layer that is crucial to achieving a smooth metal surface and meanwhile promoting electrolyte reduction and formation of a primary $M_A^+$-rich SEI. The cathodic current density for plating is kept low, which allows $M_A$ to deposit back slowly to smooth out the surface and SEI formation to complete at the same time.

As revealed by atomic force microscopy (AFM) images, large-scale close to atomic-flatness surfaces were created for the three alkali metals after application of the ESP processes. The flat terraces can reach a size of as large as ~30 μm wide with roughness of only ~1.5 nm, approaching the quality of single crystalline

surfaces (Fig. 2b–h), in contrast to the rough pristine surfaces with height variation of ~1 μm (Fig. 2a and Supplementary Fig. 2a, b). Remarkably, the SEI films of the three alkali metals are also ultra-smooth with molecular-scale roughness of e.g., ~0.6 nm within an area of 5 × 5 μm$^2$ on the metal surface (Fig. 1d, f, h). The large-scale flat surfaces can also be obtained on Li and Na thin films introduced onto modified Cu substrates (Supplementary Fig. 3), which facilitates easy characterizations and enables battery performance tests with reasonable DOD.

**Tunable structure and mechanical property of SEI**. To probe the vertical structure and mechanical property[39–41] of the ultra-smooth SEIs, AFM force curve measurements were performed, mainly on Li anodes. For convenience of discussion, we begin with inspection of the force-piezo displacement curve recorded for soaked Li surface (Fig. 3a and Supplementary Fig. 5a, d). It can be seen intuitively that the cantilever deflects steeply until the tip reaches the Li substrate so that only a dip is seen. This implies the SEI on the soaked sample is rather stiff and lacks of flexibility against tip pressing so that cracking occurs upon force accumulation, as discerned by the fragmentation of surface after the measurements (Fig. 3b and Supplementary Fig. 6a, d).

However, the situation is totally different for the polished surfaces. A single potential step into the potential region preferential for reduction of 1,3-dioxolane (DOL) in the stripping process can create an SEI with coupled rigidity and elasticity. This can be viewed with linear deflection of cantilever followed by levering off until reaching the Li substrate (Fig. 3d and Supplementary Fig. 5b, d). This behavior is just like an elastic-plastic deformation process and suggests a soft inner layer and a stiff outer layer, i.e., soft-stiff SEI structure, which is robust against pressing as discerned by the smooth surface after force curve measurements (Fig. 3e and Supplementary Fig. 6b, e). By

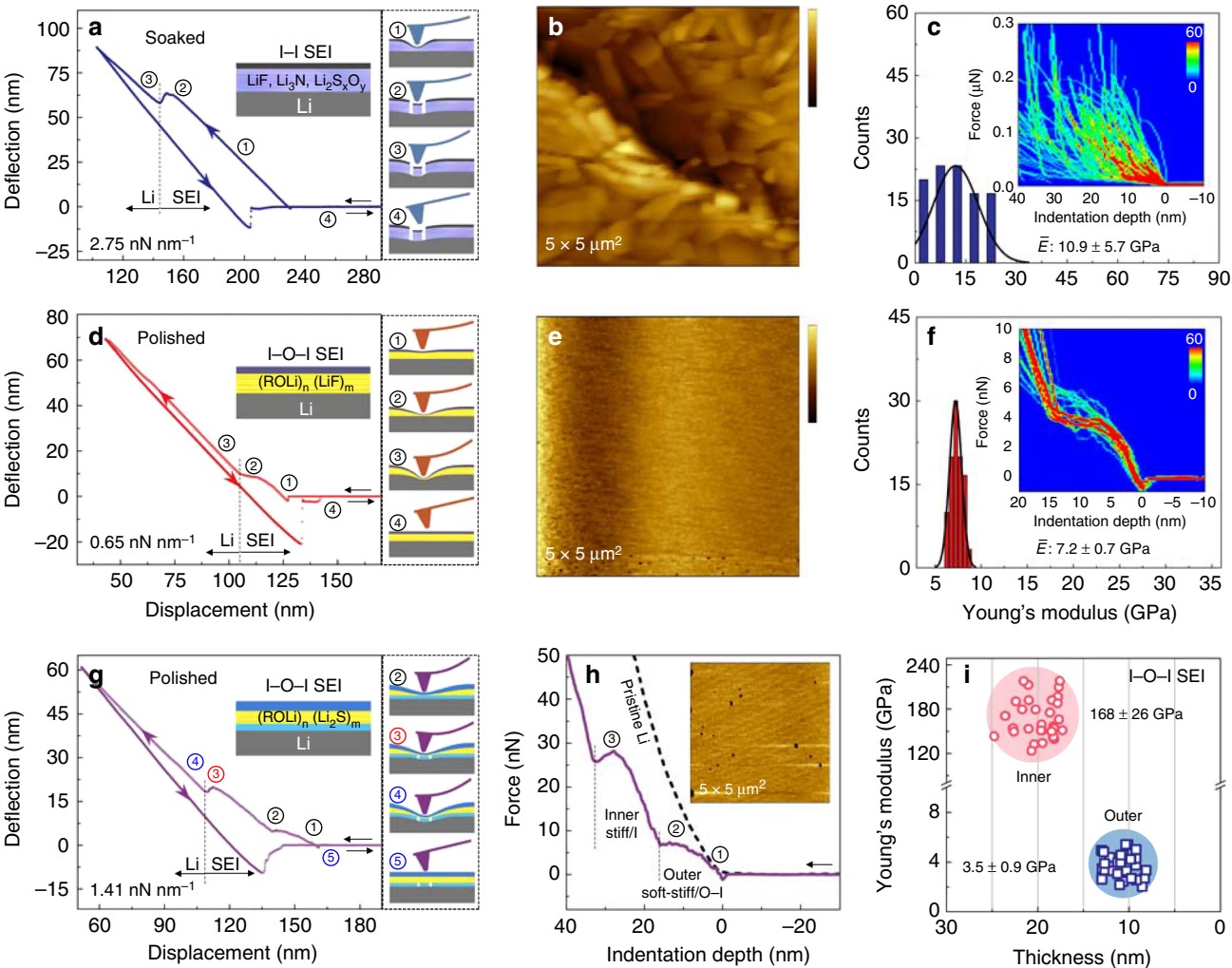

**Fig. 3** AFM characterization of mechanical properties of the SEI layers. **a** Typical force-displacement curve of soaked Li surface. **b** AFM image for soaked Li sample after the force curve measurements. Color bar, 0–0.5 μm. **c** The corresponding histograms of Young's modulus based on 80 force curves with bivariate histograms of force-indentation curves (insets). **d** Typical force-displacement of polished Li surface with single potential step in the ESP process (O–I structured SEI). **e** The corresponding AFM image after the force curves measurements. Color bar, 0–1 nm. **f** The corresponding histograms of Young's modulus based on 80 force curves with bivariate histograms of force-indentation curves (insets). **g** Typical force-displacement of polished Li surface with multiple potential steps in the ESP process (I–O–I structured SEI), showing an elastic-plastic deformation-like process followed by an elastic-cracking process. **h** Typical force-indentation curve showing an O–I combined outer layer and an I-rich inner layer. AFM image after the force curve measurements is given as an inset. Color bar, 0–2 nm. **i** Young's modulus of different regions for I–O–I structured SEI-based on 30 force curves

applying multiple potential steps sequentially to the regions for bis(trifluoromethanesulfonyl)imide (TFSI) reduction at lower potential and for DOL reduction at higher potential and then back to the region for reduction of TFSI, two types of force curves are observed for such an SEI, either with two elastic-plastic deformation-like processes (Supplementary Fig. 5c, d) or an elastic-plastic deformation-like process in the outer region followed by an elastic-cracking process in the inner region (Fig. 3g, h and Supplementary Fig. 5c, d). Both situation indicating the presence of an additional stiff inner layer, i.e., an alternating stiff-soft-stiff multi-layered SEI has been formed.

To understand the origin of the different mechanical behaviors of the three types of SEIs, depth-dependent characterization of chemical composition was performed by X-ray photoelectron spectroscopy (XPS) profile analysis. For soaked sample (Fig. 4a, b and Supplementary Fig. 8a), inorganic species of LiF, $LiNSO_2CF_3$, and $Li_3N$ are the major components in the inner region, while $LiCF_3$, $LiNSO_2CF_3$, and $Li_2S_2O_4$ in the surface region. Only very little organic moieties exist, mostly from high-oxidation state

reduction products such as $CH_3OCH_2CH_2OLi$ as the C 1s signal is weak, broad and random, largely from the aliphatic C; this is further corroborated by the broad signals of amorphous organic salts of Li observed by X-ray absorption near-edge structure (XANES)[42] (Supplementary Fig. 10). There is a cut of signals on the spectra between etching time of 30 and 60 s, dividing the SEI film into two inorganic-rich layers, which is denoted as an I–I structure or all-inorganic structure (Supplementary Fig. 9a). Interestingly, such I–I type of SEI structure is in contrast to the traditional I–O double layer model of SEI[34,43], a picture that has been widely adopted for SEI for a long time.

Next, the soft-stiff SEI formed after ESP process with a single potential step for stripping displays a dramatically different chemical structure. No passive layer is present as no signals from $Li_2O$ and carbonyl species ($CO_3^{2-}$) are observed by XPS and XANES (Supplementary Figs. 8b and 10) and no signal from hydroxyl (–OH) by Fourier transform infrared spectroscopy (FTIR) (Supplementary Fig. 11). This leaves with a fresh Li surface for SEI formation. Remarkably, however, in the inner

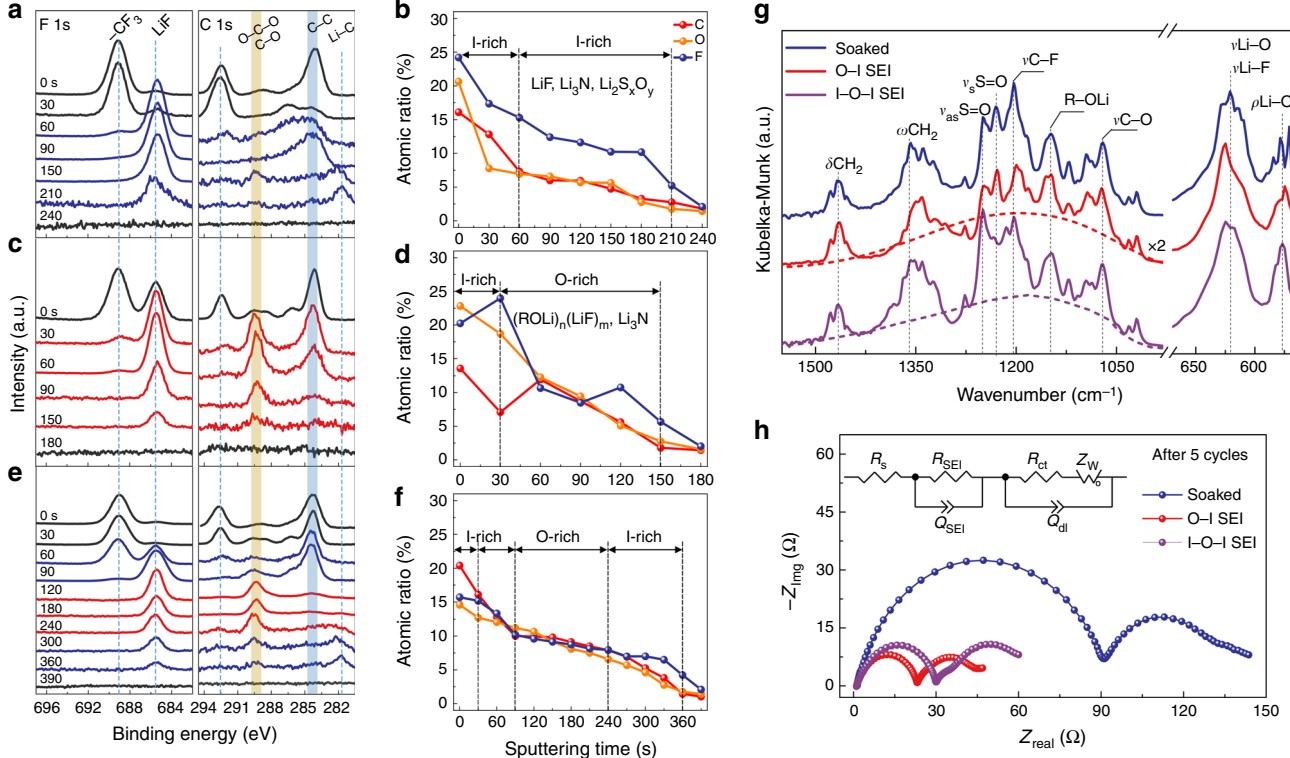

**Fig. 4** Characterization of chemical composition and structure of SEI layers. **a**, **c**, **e** XPS spectra of F and C elements recorded after different time of argon ion sputtering at soaked Li electrode (**a**) and polished Li electrodes with O–I (**c**) and I–O–I (**e**) structured SEIs, respectively. **b**, **d**, **f** The corresponding depth profiles of the atomic concentration of C, O, and F elements. **g** FTIR spectra recorded for Li electrodes of various conditions. **h** EIS measurements of soaked and polished Li electrodes taken after five cycles of galvanostatic cycling at 0.5 mA cm⁻². The proposed equivalent circuit for fitting the EIS spectra is given as an inset

region (sputtering time between 30 and 150 s), the signal of C–O moieties clearly indicates the presence of an ordered low oxidation state polymeric ROLi that is rich of C–O–C, C–C–O, O–C–O, in contrast to the almost absence of such moieties in the I–I structured SEI of soaked sample. In the surface region, however, the high-oxidation state ROLi is present judged from the appearance of aliphatic C signal; meanwhile, inorganic LiCF₃ and LiF become dominant with small amount of Li₂NSO₂CF₃ and Li₂S₂O₄ and Li₂SO₃ (Fig. 4c, d and Supplementary Fig. 8b). These results suggest an SEI structure with an organic-rich inner layer of ca. 10 nm thick, and an inorganic-rich outer layer of ca. 2 nm thick, estimated from the calibrated sputtering rate of 4 nm per min in Si, and we thus denote the soft-stiff SEI as O–I structured SEI (Supplementary Fig. 9b).

Finally, the stiff-soft-stiff SEI created with multiple potential step for stripping in the ESP process leads to an I–O–I structured SEI, in which an organic-mixed middle layer is sandwiched by two inorganic-rich layers in the inner and outer regions, respectively (Supplementary Fig. 9c), with total SEI thickness of ca. 26 nm. Note that each layer of the I–O–I structured SEI contains components and structure that have subtle differences from those of O–I structured SEI (Fig. 4e, f): The I-rich inner layer contains Li–C moiety as a result of exposure of freshly generated Li surface upon Li dissolution; the O-mixed middle layer contains C–O–C moiety with negligible aliphatic C signal; and the outmost region of the I-rich outer layer is rich of LiNSO₂CF₃ while the LiF components are buried underneath. Noteworthy is that the S-based signals also indicate the presence of substantial amount of Li₂S throughout the thickness of the SEI (Supplementary Fig. 8c).

Thus, the chemical composition and structure of the three SEIs revealed by XPS profile analysis and the mechanical properties

probed by AFM force curve measurements are corroborative of each other: The all-inorganic SEI is stiff, while the O–I and I–O–I structured SEI have coupled elasticity (soft) and rigidity (stiff). In particular, the combination of two chemically different O–I layers behave mechanically like a cushion layer that can resist against dendrite growth. The apparent Young's moduli are 7.2 ± 0.7 GPa for the O–I SEI and 3.5 ± 0.9 GPa for the combined O–I part of the I–O–I SEI. These values are smaller than that of the all–I SEI (10.9 ± 5.7 GPa), but are within the scope for rigid inorganic materials[41,44,45] (Fig. 3c, f, i and Supplementary Note 3). Statistical analysis on thickness reveal total thickness of 22 ± 10 , 11 ± 2 , and 21 ± 3 nm for the all–I SEI, O–I SEI and I–O–I SEI, respectively, after correction of Li deformation (Supplementary Note 4 and Supplementary Fig. 5e). These values are on the smaller side given the reason that the organic-rich layer is deformed elastically and not penetrated upon pressing. The coupled elasticity and rigidity presented by the ultra-thin ultra-smooth I–O–I SEI are similar to the favorable mechanical property possessed by much thicker polymeric artificial SEIs that have been demonstrated to accommodate the interfacial fluctuation during the Li plating/stripping processes[46–48].

**Mₐ⁺-rich SEI for enhanced conductivity**. The exceedingly high Mₐ⁺ surface concentration generated upon stripping of Mₐ surface favors the reduction of TFSI or trifluoromethanesulfonate (OTf) anion to low oxidation states; in the case of Li electrode, it also provides the possibility for DOL to reduce and form Li-incorporated oligomer of (ROLi)ₙ, rather than only the Li-ending ROLi chain, where R denotes for polymeric (CH₃CH₂OCH₂O–)ₙ or (CH₃OCH₂CH₂O–)ₙ moiety. Thus, it is feasible to create Mₐ-rich SEIs in all cases, and a network-like framework incorporated with

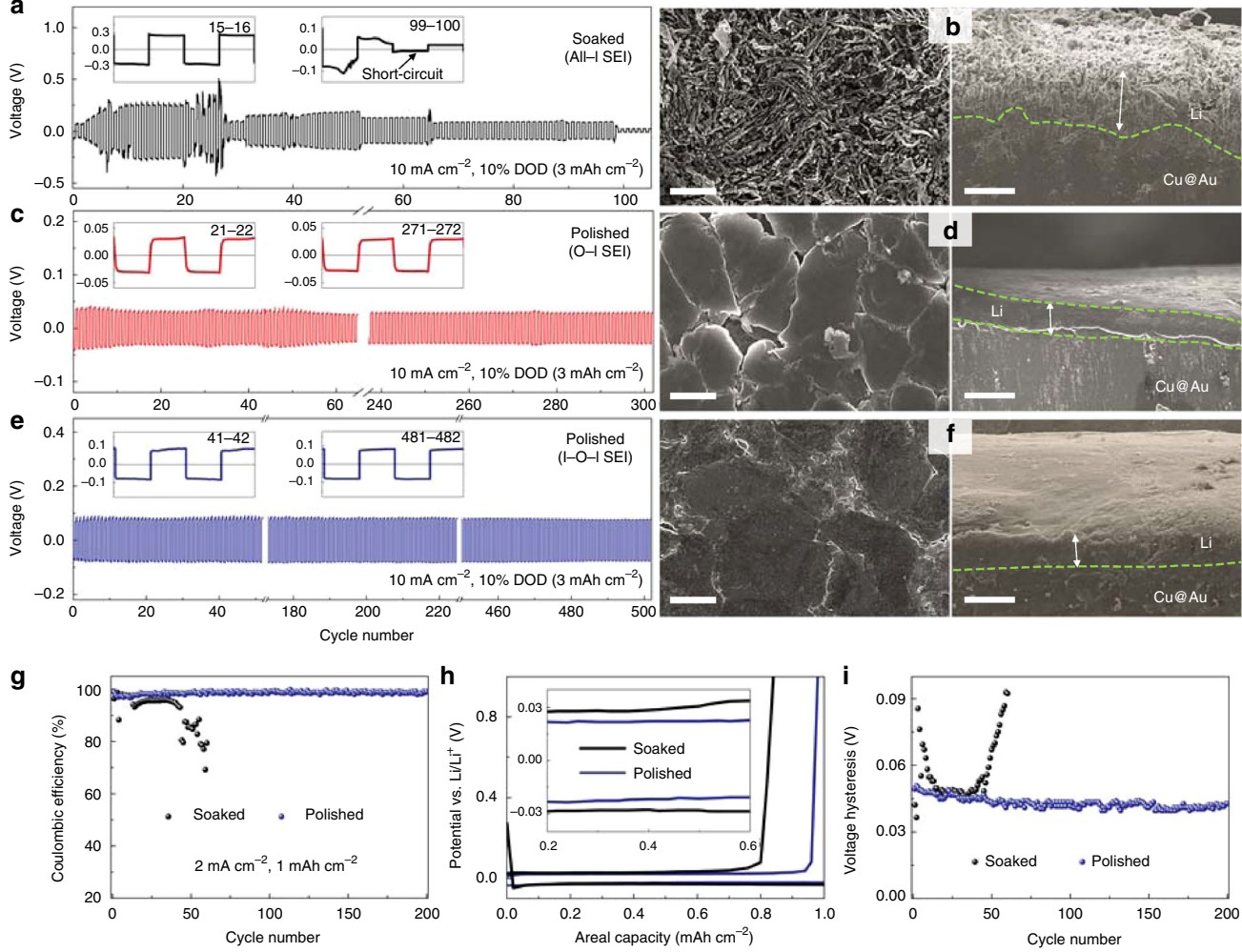

**Fig. 5** Electrochemical performance of polished and soaked Li anodes. **a, c, e** Voltage profiles of symmetric Li cells with soaked or polished Li thin-film electrodes using electrolyte of 1 M LiTFSI/DME-DOL. **b, d, f** SEM images of soaked Li electrode (**b**) and polished Li electrodes with O–I (**d**) and I–O–I structured SEI (**f**) after 100, 100 and 400 cycles, respectively. Scale bars are 10 μm (left images of **b, d, f**) and 50 μm (right images of **b, d, f**). **g–i** Coulombic efficiency (**g**), voltage profiles (**h**) and voltage hysteresis (**i**) of Cu||Li cells at 2 mA cm⁻². Both the Cu substrate and Li foil were covered by the as-prepared I–O–I SEIs

the polymeric $(CH_3CH_2OCH_2O-)_n$ or $(CH_3OCH_2CH_2O-)_n$ moiety in the case of Li electrode. In the following, we focus on Li electrode for further discussion on the SEI structure.

FTIR measurements disclose differences among the three types of SEIs (Fig. 4g and Supplementary Table 4), which supports the $Li^+$-rich network-like structure of the O–I and I–O–I SEIs. First, the signal of band at 575 cm⁻¹, attributed to the rocking mode of vibration of Li–O, increases in a sequence of I–O–I SEI > O–I SEI > all–I SEI, implying the amount of $Li^+$ in the same sequence. Second, the overwhelmingly strong bands at ~625 cm⁻¹, largely contributed by various stretching modes of the LiF clusters[49], reveal that all the three types of SEIs contain substantial amount of LiF clusters. Furthermore, the broad background-like feature in the region of 1000–1400 cm⁻¹ from C-based bands of organic components (e.g. C–C, C–O, and C–H) observed on O–I and I–O–I structured SEIs is an indication of presence of a cross-linked structure in these SEIs. These features suggest that $Li^+$-rich oligmeric $(ROLi)_n$ moieties, more preferential than the Li-ending chain-like ROLi moiety, are formed in the I–O–I and O–I SEI (see Supplementary Note 5 and Supplementary Fig. 13). Hence, a network structure composed of oligmeric $(ROLi)_n$ incorporated

with small inorganic molecules and clusters such as LiF and $Li_2S$ is proposed (see Supplementary Note 6), in which the rich amount of Li sites in the network is expected to provide channels that facilitate Li ion conduction. A conclusion can be reached based on the above analysis that the O–I and I–O–I types of SEI structures are not only on contrary to the I–I structure of the SEI of the soaked sample and disordered mosaic type of model for SEI, but also expected to exhibit fast transport of $Li^+$ ions.

Indeed, the electrochemical impedance spectroscopy (EIS) measurements (Fig. 4h and Supplementary Table 5) show that the O–I and I–O–I types of SEIs have considerably smaller SEI resistance (21 and 28 Ω) as well as charge transfer resistance (23 and 30 Ω), compared with those of soaked sample (87 and 50 Ω), respectively. This illustrates that not only the $Li^+$ transport through the thin SEI layer, but also the electron transfer across the Li–SEI interface, are significantly enhanced in the SEIs formed on the fresh and flat Li surfaces. Furthermore, given the thickness of 12 and 26 nm, conductivity of $3 \times 10^{-8}$ and $5 \times 10^{-8}$ S cm⁻¹ are estimated for the O–I and I–O–I SEIs, respectively, which are of about ten times enhancement compared to those reported for SEI[50].

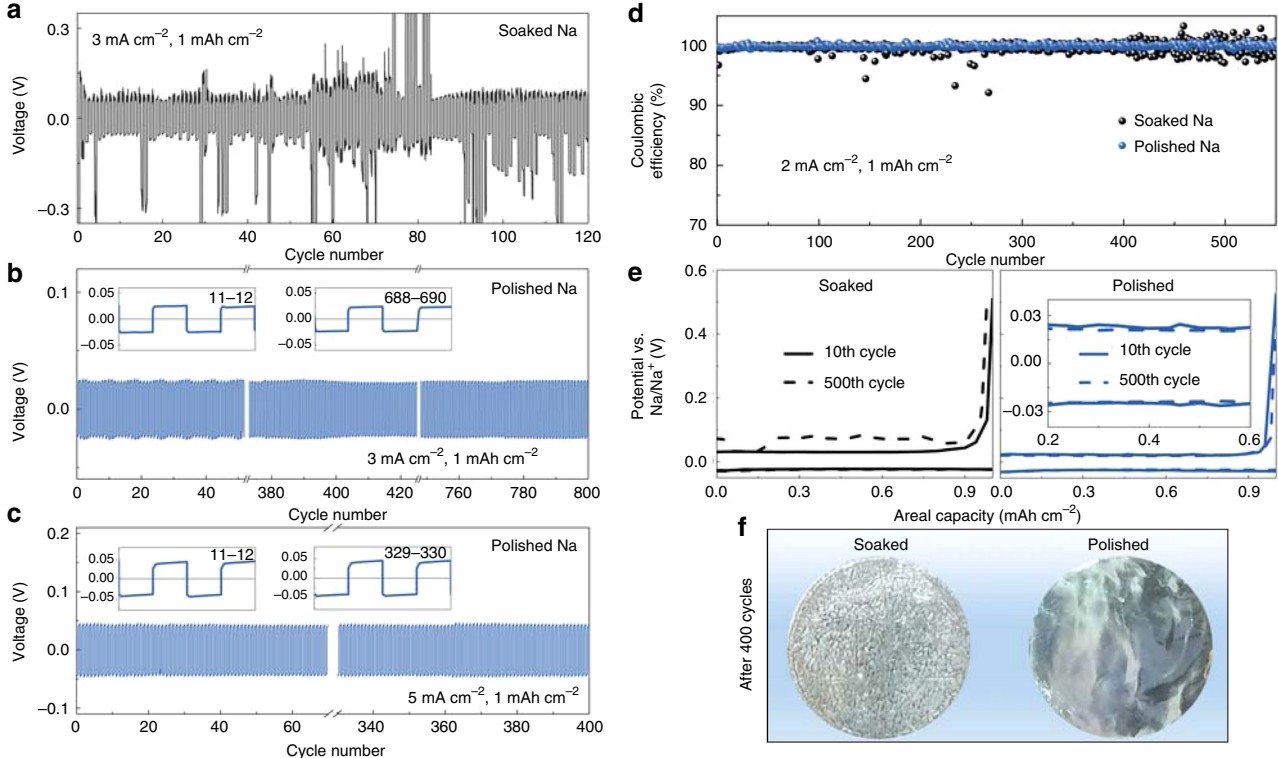

**Fig. 6** Electrochemical performance of polished Na and K anodes. **a–c** Voltage profiles of symmetric Na cells with soaked (**a**) and polished (**b**, **c**) Na foil electrodes using electrolyte of 1 M NaOTf in diglyme. **d**, **e** Coulombic efficiency (**d**) and voltage profiles (**e**) of asymmetric Cu||Na cells using electrolyte of 1 M NaOTf in diglyme at a current density of 2 mA cm$^{-2}$. Both of the Cu and Na electrodes were covered with SEIs created by the ESP process (see Methods section). **f** The optical images of soaked and polished Na foils after 400 cycles

**Enhanced electrochemical performance of polished Li anodes**. The Li-rich USUT SEI with the O–I and I–O–I multi-layered structures on flat Li surface bear all features for an ideal SEI. Of particular significance is the coupled rigidity and elasticity of these SEIs, which is a highly desirable mechanical property for stabilizing Li anodes (Fig. 5 and Supplementary Figs. 14–18). With these features and merits, Li thin-film anodes with the I–O–I SEIs can run over 500 and 300 stable cycles with 10% DOD (Fig. 5e) and 50% DOD (Supplementary Fig. 16c), respectively, at 10 mA cm$^{-2}$. To separately evaluate the capability of the as-prepared I–O–I SEI for working on a foreign substrate, asymmetric Cu||Li cells were employed (see Methods section). As shown in Fig. 5g, the Cu||Li cell can run for over 200 cycles with 100% DOD under current density of 2 mA cm$^{-2}$ and areal capacity of 1 mAh cm$^{-2}$ with an average Coulombic efficiency of ~99% in an electrolyte with ordinary concentration of Li salt (1 M LiTFSI/DME-DOL) and without additives, approaching the performance achieved by employing high Li salt concentration and cycling protocol[51].

Optimizing the ESP process can further enhance the performance of M$_A$ anodes for specific purposes. To accommodate Li–S batteries, additional 2 wt.% LiNO$_3$ was added to the electrolyte of 1 M LiTFSI dissolved in 1,2-dimethoxyethane (DME) and DOL for polishing Li electrodes. The I–O–I SEI thus prepared significantly enhances the capability to withstand the reaction of Li surface with sulfur so that at least 450 cycles was maintained in the absence of LiNO$_3$ with Li DOD of as high as 30% (see Methods section for the detailed DOD calculation) and Coulombic efficiency of nearly 100% (Supplementary Fig. 19). To further probe the potential application of the SEIs prepared in DOL-based electrolyte in Li batteries involving carbonate-based electrolyte, Li||LiCoO$_2$ full cells in the electrolyte of 1 M LiPF$_6$ in EC-DMC-EMC ($V/V/V$, 1/1/1) were constructed. The full cell using Li anode coated with I–O–I

structured SEI exhibits a promising reversibility with Columbic efficiency of 99.7% and capacity retention of 83% after 200 cycles (Supplementary Fig. 20).

The above results demonstrate that despite of the large Young's modulus of the SEI, the coarse all-inorganic structured SEI promotes dendrite growth; on contrary, the USUT multi-layered SEI with coupled rigidity and elasticity, even in the free-standing like state on Cu substrate, can suppress dendrite growth for long-term stability of Li anodes.

**Enhanced stability of polished Na and K anodes**. Na and K anodes created by ESP processes have all-inorganic type of SEI structure, which has been confirmed by the results of XPS, FTIR, and AFM (Supplementary Figs. 21–23), due to the reason that the currently adopted electrolytes compatible with Na and K anodes, NaOTf/diglyme and KTFSI/DME, respectively, do not contain DOL that would otherwise be an essence for creating organic-rich moieties. Nevertheless, compared with the all-inorganic SEI on the soaked Li anode, these SEIs are thinner, ca. 15 and 7 nm with apparent Young's moduli of ca. 1.3 and 14.2 GPa for Na and K anodes, respectively. Importantly, however, since the primary problems for Na and K anodes arise from the poor quality of SEIs which then induces dendrite growth and causes excessive electrolyte consumption, the compact and stable Na$^+$-rich and K$^+$-rich SEIs on smooth Na and K surfaces may be adequate to circumvent the problems. Hitherto unreported cycling stability was achieved for Na anodes with 100% DOD for the Cu||Na cell, which can run for at least 550 cycles with Columbic efficiency close to 100% (Fig. 6d, e). Optical images show that the surface of polished Na remains metallic luster and relatively flat after 400 cycles (Fig. 6f). As for K anodes, symmetric cells were assembled with two polished K foil electrodes. A preliminary try of the ESP process leads to enhanced

stability of at least 200 cycles (0.1 mA cm$^{-2}$ with 0.02 mAh cm$^{-2}$ in 1 M KTFSI/DME) (Supplementary Fig. 24).

## Discussion

In conclusion, we have established a simple, but more than simple, ESP-based electrochemical approach to create near-perfect M$_A$ anode with near-ideal SEIs of the three alkali metal anodes. The methodology for lateral as well as vertical characterization of structure and property of the USUT SEI by AFM-based imaging and force spectroscopy and XPS depth profile analysis provide corroborative evidence that elucidate novel well-defined O–I and I–O–I types of multi-layered SEI structure. Such Li anodes have exhibited significantly enhanced cycling stability under high-current density with high DOD, and allow Li–S battery to operate in the absence of LiNO$_3$ additive under high DOD. Also, such Li anodes with SEIs prepared in DOL-based electrolyte could be extended to match LiCoO$_2$ cathodes in carbonate-based electrolytes. The prolonged cycling stability of Na anode with 100% DOD provides solid foundation for its practical applications. Our work illustrates that an ultra-smooth and ultra-thin SEI of near-perfect alkali metal anodes may be robust enough to protect the M$_A$ anodes for superior long-term cycling stability. The near-perfect alkali metal surfaces also provide tremendous opportunities in diverse fields including surface science, nanotechnology, and materials and energy sciences and technologies.

## Methods

**Electrochemical polishing and SEI formation of M$_A$ metal foil surface**. The electrochemical polishing of M$_A$ anodes, employing a stripping-plating strategy, was conducted in a two-electrode cell that employed a larger alkali metal foil as the counter electrode before half cells or full cells were assembled. To facilitate simultaneous creation of flat M$_A$ surface and formation of smooth SEI, a potential which is sufficiently anodic of M$_A$ equilibrium potential yet within the potential window for electrolyte reduction was applied in the stripping step, during which M$_A$ dissolution and electrolyte reduction occurred concurrently. For Li anode, the potential range for stripping was chosen to be between 0.6 and 1.4 V in DOL-DME-LiTFSI electrolyte (Supplementary Fig. 1) and lasted for a period of 90 – 150 s to obtain a flat Li surface, which meanwhile supplies high concentration of Li$^+$ near the surface to facilitate the initial stage of SEI formation. Multiple potential steps were applied between 0.6 and 1.0 V (vs. Li/Li$^+$) to create multi-layered SEI. A galvanostatic cathodic polarization at low-current density is immediately followed as the plating step for a prolonged time, during which Li deposition and electrolyte reduction take place, which further smoothens the Li surface and completes the SEI formation. For Na and K anodes the stripping potentials were 0.8 V (vs. Na/Na$^+$) and 0.2 V (vs. K/K$^+$), respectively, which was followed by the plating step.

**Electrochemical polishing of Li and Na thin-film electrodes**. To test the electrochemical performance of Li and Na anodes with practical concern, thin film electrodes were prepared by a modified lithophilic or sodiophilic approach initially reported by the group of Yi Cui[52]. The procedure is described as follows: Cu foils (Φ13.0 × 0.024 mm, ≥99.7%, Power long) were washed by immersing in 0.1 M sodium oxalate for 5 min, followed by successive rinsing with deionized water, acetone and ethanol. After drying, the Cu foils were transferred to a thin film deposition platform (Explorer-14, Denton Vacuum) and sputtered with 50 nm Au to form lithophilic or sodiophilic Cu@Au foils. The Cu@Au foils were then transferred to Ar-filled glove box for Li or Na infiltration by dipping in a Li or Na melt obtained by heating a piece of metallic Li to over 250 °C or metallic Na to over 150 °C, until the melt was entirely entrapped. After cooling down to room temperature, Li or Na thin-film electrodes were obtained. The amount of the Li or Na melt was quantified depending on purposes. The Li or Na thin-film electrodes were subject to electrochemical polishing, following the same procedure for Li or Na foils, to obtain flat metal surfaces and to achieve desirable SEIs. To obtain free-standing like SEIs on modified Cu substrates for constructing anode-free type of Cu||Li and Cu||Na cells, respectively, the residual Li or Na on the corresponding thin film electrodes after electrochemical polishing were removed by electrochemical dissolution.

**Measurements of electrochemical behaviors of alkali electrodes**. Galvanostatic cycling and cathodic polarization behaviors were examined on LANHE CT2001A battery testing system (LAND Electronics) or VMP-300 multi Potentiostat (Bio-Logic Science). Both metal foil electrodes and metal thin-film electrodes were employed for evaluation of charge–discharge cycling behavior as well as the behavior against prolonged unidirectional galvanostatic polarization. For Li metal electrodes, Li foils were assembled into symmetric coin cells (CR2025-type) with soaked Celgard-2400 separator and 1 M LiTFSI/DME-DOL (1/1, V/V) electrolyte, or otherwise indicated. For galvanostatic cycling using Li thin-film electrodes at

fixed DOD, typical 10 ± 10% mg of Li was entrapped onto Cu@Au foil (1.3 cm$^2$), which can provide a total capacity of 38.6 mAh, based on the consideration of theoretical specific capacity of 3860 mAh g$^{-1}$ for metallic Li. Cycling at a current density of 10 mA cm$^{-2}$ (3 mAh cm$^{-2}$) results in an charge–discharge capacity of 3 mAh cm$^{-2}$ × 1.3 cm$^2$ = 3.9 mAh, which is equivalent to 10% DOD. Cycling tests with Cu||Li cells were carried out by first depositing 1 mAh of Li onto the Cu@Au surface with free-standing SEI, followed by Li stripping up to 1.0 V vs. Li/Li$^+$. For Na and K anodes, the configurations of coin cells were almost the same as Li anodes except the glass fiber separator. The electrolyte for Na-based cells and K-based cells are 1 M NaOTf in diglyme and 1 M KTFSI in DME, respectively.

**Measurements of performances of Li–S batteries**. Galvanostatic charge–discharge cycling was performed on LANHE CT2001A battery testing system. Li thin-film electrodes prepared by the lithophilic approach were used as the anodes for Li–S coin cells (CR2016-type). The amount of the metallic Li for entrapping was based on the actual loading of active materials for cathodes. For an example, the weight of Li was calculated according to the stoichiometric ratio of Li and S of the reaction 2Li + S → Li$_2$S. The actual DOD for Li thin-film electrodes was 30%. The metal–organic-framework/sulfur composite[5] was employed as cathode material. The composite contains a three-dimensional porous cobalt and nitrogen-doped graphitic carbon with 70 wt% sulfur loading (S@Co–N-GC). The cathode was prepared by mixing Co–N-GC/S composite powder, acetylene black, water-soluble polymer n-lauryl acrylate (LA) with a weight ratio of 7:2:1 onto Al foils and then dried at 60 °C under vacuum overnight. The electrolyte for battery testing was 1 M LiTFSI/DME-DOL (1/1, V/V), without any additives including LiNO$_3$. Cycling was performed in the voltage range of 1.7–2.7 V. The geometric areas of all electrodes were 1.3 cm$^2$ (13 mm in diameters) and the areal mass loading on the electrode was about 1.5 mg cm$^{-2}$.

**Measurements of performances of Li||LiCoO$_2$ batteries**. To test the potential application of polished Li anodes, having SEI prepared in the DOL-based electrolyte, in full cells in carbonate-based electrolytes, LiCoO$_2$ (Power long) was used as a cathode material. The LiCoO$_2$ cathode was prepared by mixing the active material LiCoO$_2$, super carbon, and LA at a weight ratio of 8:1:1 in deionized water and ethanol to form a slurry. The slurry was pasted on Al foil and dried at 60 °C under vacuum overnight. The electrolyte for battery testing was 1 M LiPF$_6$/EC-DMC-EMC (1/1/1, V/V/V). The electrode were 1.3 cm$^2$ (13 mm in diameters) with LiCoO$_2$ loading mass of approximately 6 mg cm$^{-2}$, corresponding to the areal capacity of 1 mAh cm$^{-2}$. Cells were cycled at 0.5 C (calculated based on the LiCoO2 cathode) in the potential range of 2.5–4.3 V.

**Data availability**. The data that support the plots within this paper and other finding of this study are available from the corresponding author upon reasonable request.

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

## Acknowledgements

This work was supported by the MOST projects (2015CB251102, 2012CB932902) and the NSFC projects (U1305246, 21621091, 21473147, 21533006, 21673193). We thank the XAFS station (BL14W1) of the Shanghai Synchrotron Radiation Facility. We are grateful to Profs. Z.Q. Tian, H.X. Yang, X.P. Ai, N.F. Zheng, H. Li, Y.G. Guo, J.B. Zhao, and M.S. Chen for helpful discussions and advices, Dr. X.B. Lian and Mr. X.L. Li for experimental assistance, and Mr. Y. Tian for help in graphic editing.

## Author contributions

B.-W.M., Q.-F.D., and Y.G. conceived and designed the experiments, analyzed the results, and participated in writing the manuscript. W.-W.W. and Y.G. carried out AFM measurements. Y.G. and Q.-H.W. conducted XPS experiments. Y.G. and S.T. conducted FTIR experiments. Y.-J.L., W.-Q.H., and Y.G. conducted battery performance measurements. D.-Y.W. and Y.F. contributed to calculation and interpretation of IR spectra. J.-W.Y., M.-S.Z., C.-H.F., and Q.-H.Z. contributed to discussions and interpretation of results and Z.-B.C. to technical assistance.

## Additional information

**Competing interests:** The authors declare no competing interests.

