## [Peer Review File · Nature Communications]

Reviewers' comments:

Reviewer #1 (Remarks to the Author):

This manuscript reports on designable ultra-smooth ultra-thin (USUT) solid-electrolyte interphases for Li, Na and K metal anodes based on a potentiostatic stripping and galvanostatic plating method for polishing the metal and formation of SEI. The article is well presented, clearly written and results are original and rigorously discussed. The topic is interesting to the battery community in general as metal are not only used for full cell assembly but also in half cell configuration where the metal anode is used as a counter electrode. I therefore recommend acceptance of the manuscript for publication in Nature Communications provided the following minor points are addressed:

The stripping procedure is performed with a current density of about 200 mA/cm² (p.6) and is thus not appropriate for a conditioning procedure in full cell since no cathode would be able to withstand such high current density. The fact that this procedure constitutes a pretreatment of the anode before full cell can be assembled should be mentioned in the manuscript since it has important implications for application.

The authors should indicate the potential values for the potentiostatic stripping step (for Li, Na and K). All tests appear to be done using 2 electrode cells. Could the authors comment on how the polarization of the counter electrode can affect the results? Could the use of three-electrode cell provide a better control on the formation of the I-O-I type SEI?

The term galvanostatic annealing used in the manuscript for the plating step is not appropriate. In page 16 the authors mentioned that Na and K SEI are all- inorganic type (based on AFM results). The authors should provide XPS and/or FTIR data to confirm this.

Could the authors discuss how the higher SEI component solubility in the Na system (see J. Electrochem. Soc. 162 (2015) A7060 and ChemSusChem 10 (2017) 401) can affect the USUT type SEI?

In p. 14 how SEI conductivity values were measured? In p. 11 and 12 how the SEI thickness was measured?

In fig. 2 b, c and g are the black dots evidence for pitting corrosion?

Reviewer #2 (Remarks to the Author):

The authors reported an electrochemical stripping-annealing (ESA) strategy to fabricate an ultra-smooth and ultra-thin SEI with the desired structure to stabilize the Li/Na/K metal surface in the electrolyte. The tuning of the SEI structure with the different stacking of organic and inorganic layers is interesting and attractive. There are still several major concerns that should be addressed before further consideration.

1. Please give the full name of DOD and I-O-I when they first appear in the abstract for a general audience.

2. The author mentioned that the use of pre-fabricated SEI in Li metal to improve Li-S battery in absence of LiNO₃ at both abstract and conclusion. The voltage profile evolution and sulfur cathode loading should be provided to convince it can work.

3. The light blue/dark blue layers in Figure 1b are unlabeled, which is confusing and hard for readers to quickly catch the point.

4. Could the author explain why different electrolyte systems including salts and solvent were employed to polish different metal surfaces? That means is that possible to use a universal electrolyte such as $\text{Li}^+/\text{Na}^+/\text{K}^+$ FSI⁻ in DOL/DME? Please discuss the effects of using each electrolyte system, because it may affect the chemical structure and formation of the SEI.

5. The authors used AFM indentation technique to measure the SEI thickness. This is very interesting. However, there are several concerns about the model in Figure 3 and data in Figure S5. i, The determination of the crack, elastic deformation, plastic deformation is very tricky for the SEI, because it is a very complicated system. The illustration in Figure 3a makes sense to me. The I-I is very brittle and like a one-layer structure. So the deformation means the penetration of the SEI, and the deformation value correlates the SEI thickness. However, for 3b and 3g, it is hard to determine if the layer is broken or just deformed due to the presence of the organic layer. From the illustration, I think the author believes that the organic layer is just deformed and will be back when unloading the tip. In this case, the deformation may not be exactly same as the thickness. Also, it is possibly the Li metal will deform as well. Please take this into consideration as well and experimentally identify it. ii, additional experimental data needs to be provided for the AFM indentation. The AFM images of the indentation regions of the Li metal with various SEIs should be provided. These will be helpful to determine if the SEI is penetrated or not. iii, Other characterizations should be conducted if the author would like to claim the exact thickness of the entire SEI and each layer.

6. Will the thin SEI break after long cycling? The Li deposition is not ultra-flat, therefore there is still a surface morphology change and interfacial fluctuation during cycling. Can this one-time SEI tolerate this?

7. To fulfill the requirement for 4-V or high-voltage Li-metal batteries, have the authors tried to polish the metal surface using carbonate-based electrolyte?

8. Will the SEI work for high deposition amount of Li metal such as higher than 3 mAh/cm²?

Response to Reviewers

We would like to express our appreciation for the reviewers' positive response and valuable comments on our manuscript. And we are grateful to the opportunity given to clarify the concerns raised in the reviewer's report. We hope that our additional experiments and careful replies and revisions adequately address the reviewers' comments and strengthen our revised manuscript. Our responses to the comments are listed point-by-point as follows:

Reviewer #1

General comments: This manuscript reports on designable ultra-smooth ultra-thin (USUT) solid-electrolyte interphases for Li, Na and K metal anodes based on a potentiostatic stripping and galvanostatic plating method for polishing the metal and formation of SEI. The article is well presented, clearly written and results are original and rigorously discussed. The topic is interesting to the battery community in general as metal are not only used for full cell assembly but also in half cell configuration where the metal anode is used as a counter electrode. I therefore recommend acceptance of the manuscript for publication in Nature Communications provided the following minor points are addressed.

Response: We are delighted that the reviewer has noted the originality and importance of our work and recommended for publication after minor revision. We also appreciate the constructive comments which we respond to below.

Comment 1: The stripping procedure is performed with a current density of about 200 mA/cm² (p.6) and is thus not appropriate for a conditioning procedure in full cell since no cathode would be able to withstand such high current density. The fact that this procedure constitutes a pretreatment of the anode before full cell can be assembled should be mentioned in the manuscript since it has important implications for application.

Response: The reviewer is correct that the electrochemical polishing constitutes a pretreatment of the anodes before half cells or full cells are assembled. It is not a conditioning procedure in full cell, but performed in a homemade electrolytic cell that uses a larger alkali metal foil as the counter electrode. We have now clarified the point with more descriptions in Methods section of the revised manuscript (P18, L17–L20).

Comment 2: 1) The authors should indicate the potential values for the potentiostatic stripping step (for Li, Na and K). 2) All tests appear to be done using 2 electrode cells. Could the authors comment on how the polarization of the counter electrode can affect the results? Could the use of three-electrode cell provide a better control on the formation of the I-O-I type SEI?

Response: 1) The stripping potentials for Na and K anodes were 0.8 V (vs. Na/Na⁺) and 0.2 V (vs. K/K⁺), respectively. For Li anode with O-I structured SEI, the stripping potential was set to 1.0 V (vs. Li/Li⁺), while for Li anode with I-O-I structured SEI, multiple potential steps were applied between 0.6 V and 1.0 V (vs. Li/Li⁺), and we found that the best procedure is 0.6 V, then 1.0 V and finally back to 0.6 V. We have added these information in the Methods section of the revised manuscript (P19, L5–L10).

2) We thank the reviewer for this critical comment. Yes, all the electrochemical polishing procedure was done in a two-electrode cell, in which a metal foil or film electrode for polishing is the working electrode and a corresponding metal foil whose surface area is made about *three times* of that of the working electrode (WE) as the counter electrode (CE). A potentiostat, whose reference and counter electrode inputs were both connected to the metal foil counter electrode, was employed to fulfill the polishing. In this case, when an anodic stripping potential is applied to the working electrode, a corresponding cathodic plating takes place on the counter electrode simultaneously, which indeed shift the potential of the counter electrode and thus that of the working electrode. Usually, the current density of the working electrode could reach up to 200 mAcm⁻² in the initial stage, corresponding to 70 mA cm⁻² on the CE. To examine the cathodic polarization on the CE during stripping, we performed linear sweep voltammogram of Li electrode using a three electrode-cell involving three Li foil electrodes. As shown by Fig. R1–1, under the current density of 70 mA cm⁻², the overpotential is about ca. –150 mV. For comparison, we also performed electrochemical polishing using three-electrode cell so that the potential of the working electrode is well controlled. The XPS depth profile (Fig. R1–2) show that the SEI prepared in two-electrode and three-electrode cells have similar structure and chemical components, except the thickness of that prepared in two-electrode cell is slightly thinner by ca. 4 nm at the most. Therefore, the design principle for SEI does not seem to be affected significantly by the polarization of the counter electrode in two-electrode cell. Nevertheless we totally agree that the use of three-electrode cell would provide more accurate control of potential of the working electrode and a better control on the formation of the SEI.

We have added discussion as Supplementary Note 1 in the revised manuscript.

Figure R1-1 | The polarization curves of Li electrode in 1 M LiTFSI/DME-DOL (V/V, 1/1). Scan rate: 20 mV s⁻¹.

Figure R1-2 | XPS depth profile analysis of Li anode with I-O-I structured SEI prepared by two-electrode cell (upper frames) and three-electrode cell (lower frames), respectively.

Comment 3: The term galvanostatic annealing used in the manuscript for the plating step is not appropriate.

Response: The reviewer's point is well taken. The term of "galvanostatic annealing" has been

replaced by "galvanostatic plating" throughout the revised manuscript.

Comment 4: In page 16 the authors mentioned that Na and K SEI are all-inorganic type (based on AFM results). The authors should provide XPS and/or FTIR data to confirm this.

Response: We thank the reviewer for asking us to investigate in more detail the composition and structure of the USUT SEIS on polished Na and K, respectively. In light of the reviewer's suggestion, we have carried out XPS and FTIR measurements to carefully analyze the chemical components of the Na and K SEIs.

For polished Na SEI, the XPS depth profile analysis (Fig. R1–3) show that inorganic species of NaF (F 1s), Na₂SO₄ (S 2p) are the major components throughout the SEI while Na₂O (O 1s) is present only in the inner region; only very little organic moieties of aliphatic C and C–O (C 1s and O 1s) exists in the surface region. The existences of SO₄²⁻ and C–O fragments are also observed on FTIR spectrum (Fig. R1-4). The clear cut of XPS signals on the spectra between etching time of 30 and 60 s divides the SEI film into two inorganic-rich layers. Thus both the XPS and FTIR results confirm that the USUT SEI on polished Na metal (~16 nm) is all-inorganic type.

For polished K SEI, the major inorganic species include KF, K₂SO₄, K₂S and very little organic moieties of aliphatic C and C–O, as suggested by the XPS depth profile (Fig. R1–5) and FTIR spectrum (Fig. R1–6). This confirms SEI on polished K anode (~6 nm) is also all-inorganic structured.

We have added XPS and FTIR results for Na and K as Supplementary Figs. 21 and 22 in the revised manuscript.

Figure R1–3 | The XPS spectra of USUT SEI layer on polished Na anode.

Figure R1-4 | The FTIR spectrum of USUT SEI layer on polished Na anode.

Figure R1-5 | The XPS spectra of USUT SEI layer on polished K anode.

Figure R1-6 | The FTIR spectrum of USUT SEI layer on polished Na anode.

Comment 5: Could the authors discuss how the higher SEI component solubility in the Na system (see *J. Electrochem. Soc.* 162 (2015) A7060 and *ChemSusChem* 10 (2017) 401) can affect the USUT type SEI?

Response: We thank the reviewer for this comment. Indeed, the solubility of SEI component significantly affects the SEI stability. It has been reported in literature that some soluble Na-SEI components such as Na_2CO_3 and HCOONa in the carbonate-based electrolytes degraded the cycling stability of Na anode (*J. Electrochem. Soc.* 162, (2015); *ChemSusChem* 10, 401–408 (2017); *ACS Cent. Sci.* 1, 449–455 (2015)). In our work, however, NaOTf in diglyme was used as the electrolyte for electrochemical polishing and testing, and the USUT SEI formed on Na metal surface in this system are mainly composed of NaF, Na_2SO_4 and Na_2O (Please refer to Response to Comment 4). These components are lower in solubility than Na_2CO_3 and HCOONa , and enable a higher stability of Na-SEI and thus the stability of the Na anode for electrochemical cycling. It has been demonstrated that Na anodes with USUT type SEI display superior long and stable lifetime over 400 cycles at 5 mA cm^{-2} in symmetrical Na cells and high average Coulombic efficiencies of 99.9% over 550 plating–stripping cycles at 2 mA cm^{-2} with 100 % DOD in Cu||Na cells.

To further examine the stability of the components of USUT SEI, XPS measurements were performed on Na anode that was retrieved from symmetric Na cells after 150 cycles under 3 mA cm^{-2} . As shown in Fig. R1–7, the components of the SEI are almost unchanged compared to the counterpart before cycling (Please refer to Response to Comment 4). This further confirms the stability of the USUT SEI on Na.

Figure R1–7 | Characterization of chemical composition and structure of USUT SEIs layers on polished Na anodes after cycling.

Comment 6: 1) In p. 14 how SEI conductivity values were measured? 2) In p. 11 and 12 how the SEI thickness was measured?

Response: 1) The measurement and estimation of SEI conductivity were described in the Method section and the text below Supplementary Table 5. The equation $R_{SEI} = \rho l/A$ was used to estimate the resistivity of SEI layer, where l is the thickness of the SEI layer suggested by XPS results, A is the electrode surface area, which is considered unchanged during cycling, and ρ is the resistivity of the SEI. The corresponding conductivity σ of SEI is obtained from ρ .

2) The thickness given in p. 11 of the original manuscript was estimated based on XPS depth profile measurements. In these measurements, the samples were sputtered by Ar ion at ion acceleration of 2 kV and ion beam current of 2 μ A over an area of 2×2 mm², and the sputtering rate was calibrated to be 4 nm pure min using a Si film with known thickness. Thus, the sputtered thickness of material under investigation was estimated according the time of sputtering and assuming the same rate of sputtering. The thickness given in p.12 of the original manuscript was determined based on AFM indentation force curve measurements. The total thickness of the SEI is the indentation distance up to the point after which the force curve exhibits linear response of indentation.

We wish to mention that accurate determination of the ultrathin SEI layer is extremely challengeable. In the case of AFM indentation measurement, inaccuracy may arise from possible deformation of the underneath Li surface and existence of an unpenetrated layer (in the case of elastic deformation of SEI). The former can be corrected by extrapolating the background force-indentation curve. The corrected thicknesses of O-I and I-O-I SEI are 11 ± 2 nm, and 21 ± 3 nm, respectively. These values agree reasonably well with those estimated by XPS (12 and 26 nm, respectively). Please refer to Supplementary Note 4 in the revised manuscript.

Comment 7: In fig. 2 b, c and g are the black dots evidence for pitting corrosion?

Response: The black dots seen in Fig. 2b are formed due to pit corrosion during the course of electrochemical stripping, Most of these pits are repaired after electrochemical plating process as shown in Fig. 1c. Even though a few pits still exist on the polished Li surface, their surfaces have been covered by the USUT SEIs which prevent further corrosion of the areas.

Reviewer #2

The authors reported an electrochemical stripping-annealing (ESA) strategy to fabricate an ultra-smooth and ultra-thin SEI with the desired structure to stabilize the Li/Na/K metal surface in the electrolyte. The tuning of the SEI structure with the different stacking of organic and inorganic layers is interesting and attractive. There are still several major concerns that should be addressed before further consideration.

Response: We are delighted that the reviewer has found our work interesting and attractive. We also appreciate the constructive comments which we respond to below.

Comment 1: Please give the full name of DOD and I-O-I when they first appear in the abstract for a general audience.

Response: We thank the reviewer for this comment. We have provided the full name of DOD (depth of discharge) in the abstract as well as other abbreviations in the text the first time they appear.

Comment 2: The author mentioned that the use of pre-fabricated SEI in Li metal to improve Li-S battery in absence of LiNO_3 at both abstract and conclusion. The voltage profile evolution and sulfur cathode loading should be provided to convince it can work.

Response: The reviewer's useful suggestion is well taken. The voltage profiles shown below (Fig. R2-1) are added to Supplementary Fig. 18 of the revised manuscript. The sulfur cathode loading is 1.5 mg cm^{-2} , which has also been described in the Methods section in the revised manuscript (P21, L1).

Figure R2-1 | Voltage profiles of Li-S cells at 1 C rate with electrolyte of 1 M LiTFSI in

DME-DOL (V/V, 1/1) in the absence of any additives such as LiNO_3 .

Comment 3: The light blue/dark blue layers in Figure 1b are unlabeled, which is confusing and hard for readers to quickly catch the point.

Response: Thanks for the reviewer for pointing out that the missing labels for the light and dark blue layers in Fig. 1b. The layers in light blue and dark blue represent the inorganic-rich inner layer and inorganic-rich surface layer of SEI, respectively, while the layer in yellow color represent organic-rich middle layer. They constitute the multi-layered SEI. We have modified the figure legend to clarify the issue.

Comment 4: Could the author explain why different electrolyte systems including salts and solvent were employed to polish different metal surfaces? That means is that possible to use a universal electrolyte such as $\text{Li}^+/\text{Na}^+/\text{K}^+$ FSI⁻ in DOL/DME? Please discuss the effects of using each electrolyte system, because it may affect the chemical structure and formation of the SEI.

Response: We thank the reviewer for this comment. We agree that use of a universal electrolyte would extend the generality of the electrochemical polishing for different metals ($\text{Li}^+/\text{Na}^+/\text{K}^+$). And actually we had attempted to polish all the three alkali metals using a M_A^+ -TFSI⁻ containing DOL/DME electrolyte since DOL solvent could help create ordered polymeric alkoxy metal compounds endowing the SEIs with desirable mechanical property for stabilizing alkali anodes. Unfortunately, we found while the LiTFSI-containing DOL/DME electrolyte works perfect for electrochemical polishing of Li metal, the NaTFSI and KTFSI salts are incompatible with DOL/DME solvents, and the surfaces of both Na and K were covered with a layer of white and flocculent precipitates. The use of NaFSI and KFSI did not change the situation. The reasons may be that these salts (M_A TFSI and M_A FSI in DOL-DME, $\text{M}_\text{A}=\text{Na, K}$) are very hydrophilic and hard to purify, and thus the moisture and impurities contained in the electrolytes can induce the undesirable chemical polymerization of DOL. This forced us to use the electrolytes that have been found to work with Na and K metals in literature, that is NaOTf in diglyme and KTFSI in DME.

[A paragraph was redacted here]

Comment 5: The authors used AFM indentation technique to measure the SEI thickness. This is very interesting. However, there are several concerns about the model in Figure 3 and data in Figure S5.

Response: We thank the reviewer for his/her appreciation of the AFM measurement of SEI thickness.

i, The determination of the crack, elastic deformation, plastic deformation is very tricky for the SEI, because it is a very complicated system. The illustration in Figure 3a makes sense to me. The I-I is very brittle and like a one-layer structure. So the deformation means the penetration of the SEI, and the deformation value correlates the SEI thickness. However, for 3b and 3g, it is hard to determine if the layer is broken or just deformed due to the presence of the organic layer. From the illustration, I think the author believes that the organic layer is just deformed and will be back when unloading the tip. In this case, the deformation may not be exactly same as the thickness. Also, it is possibly the Li metal will deform as well. Please take this into consideration as well and experimentally identify it.

Response: We thank the reviewer for pointing out the important issues on the SEI measurement by AFM. Indeed, AFM investigation of the complicated thin SEIs on soft alkali metals is a tough task from both the experiment and data analysis points of view. Few work have been reported so far on Li/Na/K. In the present work, the O-I or I-O-I structured SEI is ultrathin and composed of even thinner multi-layers; especially the organic layer is elastic so that tip indentation does not necessarily penetrate each layer. These impose additional challenges in experiments and accurate determination of the thickness of the SEI layer.

Thanks for the reviewer's suggestion, we re-examined the force-indentation curves of pristine and polished Li surfaces for more accurate determination of the SEI thickness. As shown in Fig. R2–2a, the force-indentation curve of pristine Li appears inclined, unlike the straight upright force increase for hard sample such as Si, implying that the pristine Li surface does deform upon indentation. Similar feature can be seen on the force-indentation curve recorded for the polished Li with I-O-I type SEI in the region after the inorganic inner layer of the SEI becomes cracked (see the arrow for the cracking point), meaning that the Li deformation is involved during indentation. We therefore made corrections on thickness by extrapolating the background-like force-indentation curve into the region of SEI deformation (Fig. R2–2b), and then subtracting the indentation depth of Li from that of SEI under the same force value to remove the Li surface deformation. The corrected total thickness of the I-O-I SEI is 24 nm. For the polished Li with O-I type of the SEI, applying the same procedure for correction leads to a thickness of 13 nm of the O-I SEI. These values are slightly reduced compared with the thickness measured by XPS (12 and 26 nm for O-I and I-O-I respectively), and may be partially attributed to the presence of the unpenetrated and compressed O-I

complex (outer) layer. Despite of the discrepancy, both techniques suggest that the SEI are on polished Li surface are ultrathin.

Figure R2-2 | a–b, The force-indentation curves of pristine Li and polished Li with O-I and I-O-I structured SEI before (a) and after (b) indentation depth subtracted by deformation of Li substrate. **c,** Schematic diagrams of AFM indentation status on different type of Li anodes in the later stages.

Finally, we wish to note two points: First, the extrapolation of the force-indentation curve of Li does not enter the initial region of the force curve of the two polished types of Li electrodes. Therefore the apparent Young's moduli for the complex O-I layers in both O-I and I-O-I SEIs remain unchanged. Second, the force-indentation curves of pristine Li and the polished Li with I-O-I SEI appear similar, but are different from that of the polished Li with O-I SEI. The reason may be as follows: The surfaces of both the pristine Li and the polished Li with I-O-I are in contact with the stiff inorganic-rich layer (native layer consisting of Li_2O on pristine Li and inner inorganic layer consisting of LiF on the polished Li surface), while the polished Li surface with O-I structured SEI layer with the soft organic-rich inner layer (Fig. R2-2c). Even though the inorganic-rich inner layer cracks in the case of I-O-I SEI, it stays on the Li surface and the tip presses the Li surface through this layer. Therefore, it is the Li surfaces which are covered by the organic-rich or inorganic-rich species that are deformed, causing the observed similarity and discrepancy in the force-indentation curves among different samples.

To clarify this point, we have briefly modified text on P12, L11–L15 and added

Supplementary Note 4 for more detailed discussion.

ii, additional experimental data needs to be provided for the AFM indentation. The AFM images of the indentation regions of the Li metal with various SEIs should be provided. These will be helpful to determine if the SEI is penetrated or not.

Response: All the three types of samples after indentation measurements had been shown in Fig. 3 and Supplementary Fig. 6 of the original manuscript. It is clearly seen that the polished surface maintains intact, while the soaked surface breaks into fragments, which implying that the I-I structured SEI on the soaked Li is brittle while the O-I and I-O-I structured SEIs on the polished Li are elastic and not penetrated. Here we further provide additional small-scale ($1 \times 1 \mu\text{m}^2$) AFM images of soaked and polished surfaces before and after an array of 8 by 8 force curves were taken, Fig. R2–3. The results are consistent with previous ones and support the conclusion that the I-I structured SEI is brittle while the O-I and I-O-I structured SEIs are elastic and not penetrated.

Figure R2–3 | AFM images of polished and soaked surfaces before and after an array of 8 by 8 force curve measurements.

iii, Other characterizations should be conducted if the author would like to claim the exact thickness of the entire SEI and each layer.

Response: We thank the reviewer for this comment. Characterization of the total thickness and thickness of each layer of the multi-layered ultrathin SEI prepared in the present work is extremely challengeable. We employed two complementary techniques, i.e. XPS depth profile analysis and AFM indentation measurement to characterize the total thickness of the SEI, and value suggest by the two techniques agree well from each other. However, while the thickness of each layer can be estimated by the XPS depth profile analysis, it is hard to get information by AFM indentation technique about the exact thickness of each ultrathin layer when interference of the adjacent layers can occur. For the XPS approach, the inaccuracy arise from the difference in sputtering rate for the sample under investigation (i.e. SEI) and for the sample used as reference for calibration (i.e. Si).

Apart from AFM and XPS, time-of-flight secondary ion mass spectrometry (TOF-SIMS) and Rutherford backscattering spectrometry (RBS) are the two main techniques to study SEI. However, these two techniques have their own weaknesses. Taking the TOF-SIMS technique as an example, while it has advantages of high sensitivity, it measures the distribution of charged particles by their mass/charge values, and the interpretation of results is not straightforward, unlike for XPS and FTIR (*Energy Environ. Sci.* 10, 513–537 (2014)). It is often rather difficult to relate the charged particles to their original precursors (*Aurbach, D. Nonaqueous electrochemistry. Marcel Dekker, 1999*).

Given the situations outlined above, we believe that quantitative mechanical analysis and modeling may be an appropriate approach to combine with in order to elucidate more accurately the thickness of each layer of an ultrathin SEI. And we hope that this may be regarded as a follow-up subject for study. So we have rephrased the sentence about the thickness of each layer in the revised manuscript (P11, L9–L12).

Comment 6: Will the thin SEI break after long cycling? The Li deposition is not ultra-flat, therefore there is still a surface morphology change and interfacial fluctuation during cycling. Can this one-time SEI tolerate this?

Response: We thank the reviewer for this comment. In general Li deposition is not flat and Li dendrite may grow and be promoted by localized effects that are inherently correlated with the roughness of Li surface as well as inhomogeneity of SEI. In the present work, however, the localized effects are largely suppressed by the use of the polished Li anode with atomically-flat Li surface and M_A^+ -rich ultra-smooth ultra-thin SEI that has coupled rigidity and elasticity. The polished Li anode exhibits outstanding electrochemical performances, as shown in Fig. 5 and Supplementary Figs. 13–15 of revised manuscript.

To further demonstrate the effectiveness of ultra-thin SEI, AFM images of the surface morphologies of the Li anode after cycling to various extents under a high current density and capacity (10 mA cm^{-2} , 3 mAh cm^{-2}) are provided as shown by Fig. R2–4. It can be seen that the Li anode remains ultra-smooth with roughness of $\sim 2 \text{ nm}$ after first 200 cycles; the smoothness of the surface only slightly decrease after 400 cycles (roughness of $\sim 20 \text{ nm}$), and no dendrite is formed yet. In addition, XPS depth profiles also measured after 200 cycles under 10 mA cm^{-2} (3 mAh cm^{-2}). As shown in Fig. R2–5, the vertical structure, chemical composition and thickness of the SEI are almost unchanged.

We hope that the above supplementary results have well addressed the reviewer's concern on stability of SEI after long cycling. Meanwhile, we have now added these additional results as Supplementary Figs. 16 and 17 of the revised manuscript.

Figure R2–4 | AFM images showing the morphological changes of Li anode with I-O-I type SEI. Electrodes were harvested in the plated state from symmetric Li cells cycled at 10 mA cm^{-2} (3 mAh cm^{-2}) with 1 M LiTFSI in DME-DOL (V/V, 1/1) electrolyte after 200–400 cycles.

Figure R2–5 | XPS spectra of Li anode with I-O-I structured SEI after 200 cycles.

Comment 7: To fulfill the requirement for 4-V or high-voltage Li-metal batteries, have the

authors tried to polish the metal surface using carbonate-based electrolyte?

Response: We thank the reviewer for this comment. Actually, we have tried to polish the Li anode using electrolyte of 1 M LiPF₆ in EC-DMC-EMC (V/V/V, 1/1/1). However, in this electrolyte, we could obtain neither the atomically-flat Li metal surface, nor the ultra-smooth ultra-thin SEI (Fig. R2–6). This situation excludes carbonate-based electrolyte to be a good choice for creating near-perfect Li anodes.

[Figure R2–6 was redacted here]

Nevertheless, the SEI formed on Li metal after electrochemical polishing can be extended to match a LiCoO₂ cathode in carbonate-based electrolyte. To demonstrate its performance, Li||LiCoO₂ cells in the electrolyte of 1 M LiPF₆ in EC-DMC-DEC (V/V/V, 1/1/1) were constructed. As shown in Fig. R2–7, the full cell using the polished Li anode with I-O-I SEI (prepared in electrolyte of LiTFSI in DOL-DME) exhibits a promising reversibility with Columbic efficiency of 99.7% and capacity retention of 83% after 200 cycles. This confirms the polished Li anode with the ultrathin SEI could fulfill full cell measurement in carbonate-based electrolyte.

We have now added these results into Supporting Information (Supplementary Fig. 19) and corresponding discussion in the revised manuscript (P15–P16).

Figure R2-7 | Electrochemical performance of Li||LiCoO₂ full cell. **a**, Cycling performances at 0.5 C (calculated based on the LiCoO₂ cathode) of the full cells with pristine Li anode and Li anode polished in 1 M LiTFSI in DOL-DME (V/V, 1/1). **b,c**, The corresponding voltage profiles of different Li||LiCoO₂ full cells. The loading mass of LiCoO₂ is approximately 6 mg cm⁻², corresponding to areal capacity of 1 mAh cm⁻² and cells were cycled in the potential range of 2.5–4.3 V (vs. Li/Li⁺).

Comment 8: Will the SEI work for high deposition amount of Li metal such as higher than 3 mAh/cm²?

Response: In light of the reviewer's comment, we measured cycling stability of Cu||Li coin cell under high areal capacity. Fig. R2-8 shows the Coulombic efficiency versus cycle number of cells in 1 M LiTFSI/DME-DOL without additives. It can be seen that under areal capacity of 4 mAh cm⁻² at 2 mA cm⁻², the average Coulombic efficiency of Cu||Li cell with I-O-I structured SEI maintains ~99% for at least 150 cycles, whereas for the cell with soaked SEI the efficiency decreases to <90% after just 25 cycles. The result clearly demonstrates that the Li anode with I-O-I SEI layer can work under high areal capacity.

Figure R2-8 | Coulombic efficiency of Cu||Li cells at 4 mAh cm⁻² under 2 mA cm⁻².

Reviewers' comments:

Reviewer #1 (Remarks to the Author):

The authors answered all the referee questions and comments and the manuscript can now be considered for publication in Nature Communications

Reviewer #2 (Remarks to the Author):

The authors replied all my questions in a proper way, and most of the explanations make sense to me. A few remaining concerns in this work need to be further discussed before being published.

1. In the response to Comment 5 Fig. R2-3, the authors showed the AFM images of polished and soaked surfaces before and after the AFM indentation experiment, if my understanding was right. However, I cannot observe the trace/spots left by the AFM tip contact and please indicate the contact position between the surface and AFM tips in the figure. For the I-I SEI, there should be some holes or pits caused by the AFM tip loading since the indentation depth/deformation was larger than 50 nm. For an organic layer, it should be extremely elastic or self-healed to get an intact surface morphology after the indentation. The SEI, a multi-layered organic-inorganic composite, is hard to present such a good elasticity. Recent work in this field using polymeric materials for SEI construction should be a good example to address this issues and therefore to be cited.

2. I suggest modifying the legends of Fig. 4 to directly point out that the XPS spectra represent the depth profiling result for the convenience of readers.

3. One more suggestion is that the author can calculate the elemental concentrations of C, F, O, Li, etc. of the SEI layers based on the XPS result, which could be helpful to differentiate the organic and inorganic layers.

Response to Reviewers

Reviewer #1

The authors answered all the referee questions and comments and the manuscript can now be considered for publication in Nature Communications

Response: We appreciate the reviewer for approving the publication of our work. We also wish to thank the reviewer's constructive comments for improving the quality of our work.

Reviewer #2

The authors replied all my questions in a proper way, and most of the explanations make sense to me. A few remaining concerns in this work need to be further discussed before being published.

Response: We are delighted that the reviewer is near to satisfactory with our replies, and thank the reviewer very much for the positive assessments on our work. We welcome the additional comments from the reviewer on a few important points about this work, and have made further revisions in the manuscript. We hope the concerns from the reviewer have been carefully addressed and clarified. Our responses to the points are as follows:

Comment 1: In the response to Comment 5 Fig. R2-3, the authors showed the AFM images of polished and soaked surfaces before and after the AFM indentation experiment, if my understanding was right. However, I cannot observe the trace/spots left by the AFM tip contact and please indicate the contact position between the surface and AFM tips in the figure. For the I-I SEI, there should be some holes or pits caused by the AFM tip loading since the indentation depth/deformation was larger than 50 nm. For an organic layer, it should be extremely elastic or self-healed to get an intact surface morphology after the indentation. The SEI, a multi-layered organic-inorganic composite, is hard to present such a good elasticity. Recent work in this field using polymeric materials for SEI construction should be a good example to address this issues and therefore to be cited.

Response: We thank the reviewer for this comment. Yes, Fig. R2–3 showed the AFM images of polished and soaked surfaces before and after the AFM indentation experiments. All the AFM indentation experiments were performed in an 8 by 8 array, which is guided by an 8 by 8 grids (see Fig. R1 below), the center of each grid being the default location of each indentation process. For the soaked samples (see Supplementary Fig. 6 in the revised

manuscript and Fig. R1 below), the surface breaks into fragments after the indentation measurements due to the brittle character of the I-I structured SEI. Therefore, no holes or pits associated with the 8 by 8 grids can be directly observed. While for the polished Li samples with O-I and I-O-I structured SEIs, the surfaces still appear intact after the indentation measurements (see Supplementary Fig. 6 in the revised manuscript and Fig. R1 below). This implies that surface indentation by depth of ca. 10 nm (O-I SEI) or 20 nm (I-O-I SEI) may be recovered completely. Such a character reveals that, within the above-mentioned indentation depth, these multilayered SEIs are indeed excellent inelasticity. Nevertheless, it should also be noted that in the case of I-O-I SEI, the inorganic-rich inner layer cracks upon indentation while the inorganic-rich outer layer still deforms elastically, as indicated in the corresponding force curves (see Fig. 3g and h in the revised manuscript). This feature reveals the buffer role of the elastic organic-rich middle layer, which changes the mechanical property of the inorganic-rich outer layer.

Figure R1 | AFM images of polished and soaked surfaces before and after an array of 8 by 8 force curve measurements.

Also, we thank the reviewer for the suggestion of citing the works about SEI construction using polymeric materials. Indeed the artificial SEIs made of polymeric materials are expected to show better elasticity and thus to benefit stable cycling performance of Li anodes. Typical works about artificial SEI construction include *Nat. Commun.* 8, 850 (2017), *Adv. Mater.* 29, 1605531 (2017), and *Angew. Chem. Int. Ed.* 10.1002/anie.201710806 (2018),

in which excellent elasticity and flexibility were demonstrated. We have cited these works by mentioning the favorable mechanical property of the polymeric artificial SEI in the revised manuscript, please refer to Page 12 Line 15–18. Meanwhile, we wish to emphasize here that the formation principle as well as nature of the multi-layered I-O-I structured SEI presented in this work are in contrast to those of polymeric artificial SEI.

Comment 2: I suggest modifying the legends of Fig. 4 to directly point out that the XPS spectra represent the depth profiling result for the convenience of readers.

Response: We thank the reviewer for this comment. We have modified the figure legend of Fig. 4 in the main text to strengthen the XPS depth profile evaluation of Li electrodes to make the figure easier to follow.

Comment 3: One more suggestion is that the author can calculate the elemental concentrations of C, F, O, Li, etc. of the SEI layers based on the XPS result, which could be helpful to differentiate the organic and inorganic layers.

Response: We thank the reviewer for this suggestion. Actually, we have already calculated the atomic concentration of various elements along with sputtering time as shown in Fig. 4b, d, f and Supplementary Fig. 8d, e, f. The depth-dependent evolution of the atomic concentration suggests the organic-rich and inorganic-rich multi-layer nature of the SEIs. Detailed discussions on organic and inorganic moieties have been given together with results from FTIR and XANES.

In light of the reviewer's suggestion, we further calculated the percentage of some specified species/moieties to differentiate the type of SEIs on different Li surfaces. The results are shown in Fig. R2 below, and further aided following discussion: For the soaked sample (see Fig. R2a), inorganic species of LiF, $\text{Li}_2\text{S}_x\text{O}_y$ and Li_3N , *etc.* are the major components throughout the whole thickness range, while only very little organic moieties exist, which indicate the all-inorganic structure of soaked SEI. For the polished sample with a single potential step in the electrochemical polishing process (see Fig. R2b), the C–O moieties are present in the inner region (in the form of an ordered low oxidation state polymeric ROLi as suggested also by the results of FTIR and XANES), while inorganic species of LiF, $\text{Li}_2\text{S}_x\text{O}_y$, Li_3N , *etc.* and $\text{LiNSO}_2\text{CF}_3$ and aliphatic C are the major components in the surface region. These results suggest an SEI structure with an organic-rich inner layer and an inorganic-rich outer layer (i.e. O-I SEI). For the polished sample with multiple potential steps in the

electrochemical polishing process (see Fig. R2c), an organic-mixed middle layer sandwiched by two inorganic-rich layers in the inner and outer regions (i.e. I-O-I SEI) is discerned. The subtle differences compared to the O-I structured SEI is that substantial amount of Li_2S_n throughout the thickness of the SEI, and the outmost region of the I-rich outer layer is rich of $\text{LiNSO}_2\text{CF}_3$ while the LiF components are buried underneath.

We have added the calculated percent composition of different SEIs as Supplementary Fig. 9 in the revised manuscript.

Figure R2 | Calculated percentages of some specified compositions of SEI layers based on the XPS spectra at soaked Li surface (a) and polished Li surfaces with O-I SEI (b) and I-O-I SEI (c), respectively. For the soaked SEI, the percentage of compositions of the outer layer and the inner layer are the summations of those calculated according to the spectra recorded after 0 and 30 s and after 60~240 s sputtering, respectively. For the O-I SEI, the percentage of composition of the outer layer and the inner layer are the summations of those calculated according to the spectra recorded after 0 s and after 30~180 s sputtering, respectively. For the I-O-I SEI, the percentage of composition of the outer layer, the middle layer and the inner layer are the summations of those calculated according to the spectra recorded after 0~90 s, after 120~240 s, and after 300~390 s sputtering, respectively.